# Progress on V_2_O_5_ Cathodes for Multivalent Aqueous Batteries

**DOI:** 10.3390/ma14092310

**Published:** 2021-04-29

**Authors:** Emmanuel Karapidakis, Dimitra Vernardou

**Affiliations:** Department of Electrical and Computer Engineering, School of Engineering, Hellenic Mediterranean University, 71410 Heraklion, Greece; karapidakis@hmu.gr

**Keywords:** aqueous batteries, magnesium-ion, zinc-ion, aluminum-ion, cathode, vanadium pentoxide

## Abstract

Research efforts have been focused on developing multivalent ion batteries because they hold great promise and could be a major advancement in energy storage, since two or three times more charge per ion can be transferred as compared with lithium. However, their application is limited because of the lack of suitable cathode materials to reversibly intercalate multivalent ions. From that perspective, vanadium pentoxide is a promising cathode material because of its low toxicity, ease of synthesis, and layered structure, which provides huge possibilities for the development of energy storage devices. In this mini review, the general strategies required for the improvement of reversibility, capacity value, and stability of the cathodes is presented. The role of nanostructural morphologies, structure, and composites on the performance of vanadium pentoxide in the last five years is addressed. Finally, perspectives on future directions of the cathodes are proposed.

## 1. Introduction

The imminent global energy crisis, with increasing consumption of fossil energy and growing ecological concerns related to the greenhouse gases emitted into the atmosphere, has led to the intensive development of new energy technologies to secure energy supply for the future. To address the rapid depletion of fossil fuels and meet the growing demand for green energies (i.e., renewable energy sources, such as solar and wind energy), an effective medium to store and transfer energy is required. Thus, the need for clean and efficient energy storage has moved to the center of attention.

Among them, organic lithium ion batteries are widely used in portable devices due to their high energy density, cycle stability, and energy efficiency, leading to the unbelievable development of smartphones, personal computers, and smart grids. However, their enlargement for the electrical grid is hindered by the safety and cost, mainly of organic electrolyte and electrode material [1]. Taking this into consideration, aqueous lithium ion batteries are promising alternatives not only in the field of grid-scale energy storage, but also for wearable applications [2]. In the last case, it was found that a flexible LiVPO_4_F material can act as both cathode and anode in a “water-in-salt” gel polymer electrolyte exhibiting energy density of 141 Wh kg^−1^, with power density of 20,600 W kg^−1^, and output voltage of 2.4 V during >4000 cycles. The use of aqueous electrolytes offers tremendous competitiveness in terms of low cost, environmental benignity, and high tolerance against electrical and mechanical mishandling [3]. Nevertheless, they suffer from a limited operating window, which is <2 V due to water electrolysis. It has been indicated that this can be extended up to 3–4 V through “water-in-salt” electrolytes [2] and hydrate-melt electrolytes [4].

Lithium ion batteries may deliver both high power and energy density, but the limited lithium resource, and increased cost, safety, and environmental issues cannot be neglected [5]. From that perspective, multivalent batteries tend to increase the energy density, since multivalent ions can transfer two or three electrons per ion [6].

Among the multivalent batteries, the magnesium ion battery (MIB) is the most studied, because magnesium metal is considered an ideal anode material due to its low reduction potential (−2.37 V), low price, and high abundance in the earth’s crust [7]. Aqueous magnesium ion batteries exhibit high ionic conductivity, absence of magnesium oxide formation, and fast mobility of magnesium ions in water solution [8]. Nevertheless, in some cases the diffusion process in cathode materials is observed to be slow due to the strong electrostatic interaction between the Mg^2+^ and the host material, resulting in the absence of high-performance electrode materials [9].

Zinc batteries have also attracted attention because they present high theoretical capacity (820 mAh g^−1^), low electrochemical potential, two-electron transfer during the redox reactions leading to a high energy density, high natural abundance, low toxicity, and intrinsic safety [10,11]. Issues of dendritic zinc have almost been sorted, replacing the alkaline electrolyte with mild neutral pH.

On the other hand, aluminum ion batteries can theoretically provide three times more charge per transferred ion as compared to lithium. If successful, multivalent systems such as aluminum could be a major advancement in the storage of energy due to the high theoretical specific capacity of 2.98 Ah g^−1^, which is the second highest compared with Li (3.86 Ah g^−1^), and much higher than magnesium (2.20 Ah g^−1^) and zinc (0.82 Ah g^−1^) [12]. Besides, it suffers from corrosion in aqueous electrolytes, which leads to the formation of a passive surface layer on the electrode, reducing the material’s efficiency. In addition, it is hard to develop aqueous aluminum ion batteries due to the low reduction potential of Al^3+^ (~1.68 V vs. standard hydrogen electrode, which is lower than H^+^) in aqueous solutions. Recently, Pan et al. reported a novel aqueous cell consisting of Al metal and graphite as electrodes, and “water-in-salt” high concentration aqueous AlCl_3_ solution as electrolyte [13]. It reached a capacity of 165 mAh g^−1^ at 500 mA g^−1^, exhibiting 95% coulombic efficiency over 1000 cycles.

Nevertheless, the availability of cathode materials for Mg^2+^/Al^3+^ ions is limited, owing to the numerous interactions within the lattice of the host material and with other species present in the electrolyte, making the diffusion of such high-charge-density ions slow [14,15]. Regarding the Zn-ion batteries, there are still challenges to overcome. The greater positive charge and the slow mobility and kinetics during the intercalation process imply that identifying suitable cathodes for aqueous Zn-ion batteries is difficult compared with Li-ion batteries. Critical characteristics, such as specific capacity and cycling stability, are primarily determined by the intrinsic electrochemical properties of electrode materials, and especially cathode materials, which to a great extent determine the energy density of a battery [16]. However, the research on aqueous batteries is in continuous progress in the development of high-energy-density cathode materials. In this review, we will highlight the progressive advancements in developing vanadium pentoxide (V_2_O_5_) cathodes with their respective performances for Mg-, Zn-, and Al-ion aqueous batteries in the last five years. There are numerous advantages of V_2_O_5_ as a cathode material. It has low cost, high abundance, ease of synthesis, low toxicity, and high theoretical capacity (~437 mAh g^−1^) compared with LiCoO_2_ (~274 mAh g^−1^) [17]. Furthermore, it consists of layers that hold together by van der Waals interactions, giving excellent possibilities for the development of energy storage devices. Finally, we aim to underline the future prospects of the cathode.

## 2. Storage Mechanism

For storage systems based on Mg and Al, the intercalation/deintercalation process is based on three mechanisms: intercalation, alloying, and conversion [3]. Among the three mechanisms, intercalation holds the credit in the majority of batteries. In a typical electrochemical intercalation process, the mobile ions enter into the lattice structure of the host. Regarding the Zn-ion batteries, the reaction mechanisms are complex and unclear. However, based on the literature, the following four main reaction mechanisms have been proposed [18]: reversible Zn^2+^ insertion/extraction, chemical conversion reactions, reversible Zn^2+^ and H^+^ co-intercalation/deintercalation, and the dissolution-deposition reaction mechanism.

Cathodes undergo both electronic and ionic transfers in the electrode/electrolyte interface and the electrode bulks during the process. In the case of multivalent cations, the diffusion suffers from slow kinetics due to the high electrostatic interactions when compared to monovalent ions [6]. Among Mg^2+^ and Zn^2+^, Al^+3^ presents more kinetic issues [19] due to the following:(1)Strong electrostatic interactions.(2)Lack of feasibility to tolerate massive electrons injection resulting in lattice structural changes.(3)Stronger salvation and bonding issues taking place as compared with Mg^2+^ and Zn^2+^.(4)Proceeding of the electrochemical reactions, along with conversion path driven by thermodynamics and low reversibility.

## 3. Vanadium Oxides

Vanadium-based compounds exhibit a range of oxidation states, including V^5+^, V^4+^, V^3+^, and V^2+^, making them feasible to composite with many other anions and cations to form vanadium oxides, vanadium carbides, vanadium nitrides, vanadium sulphides, vanadium phosphates, and metal vanadates. Among them, vanadium oxides have attracted interest for energy storage in the past decades. Due to the distortion of V-O coordination polyhedral and the conversion of diverse oxidation states, vanadium oxide compounds present higher specific capacity compared with the other vanadium-based materials when they are used as electrodes [20]. The most common phases are orthorhombic V_2_O_5_, bilayered V_2_O_5_·nH_2_O, VO_2_, V_3_O_7_·H_2_O, V_6_O_13_, and V_2_O_3_. Orthorhombic V_2_O_5_ and bilayered V_2_O_5_·nH_2_O possess layered structures with open ion diffusion channels between layers [21], strengthening the electrochemical performance. Nevertheless, they show low stability in aqueous electrolytes, poor conductivity, and a low ionic diffusion coefficient, which consequently results in poor long-term cycling performance. In the following sections, the general strategies undertaken to improve the cathode performance are indicated, and the structure–performance relationship as reported in the last five years is discussed.

### 3.1. Approaches to Improve the Electrochemical Performance of V_2_O_5_

#### 3.1.1. Nanostructure Engineering

Nanostructure engineering is a promising approach to enhance the electrochemical performance of cathode materials, dealing with all aspects of design and structures on the nanoscale. Nanostructured electrode materials (tens of nanometers) offer large surface area, more active sites, and short electron/ion transport paths, which is beneficial to accelerate reaction kinetics enabling high rate capability. The nano-sized hollow structure cannot only alleviate the structural stress upon cycling, but it can also provide shortened ion and electron transport paths, and enhance the surface capacitive behavior. V_2_O_5_ nanospheres with a diameter of about 450 nm and shell thickness of 50 nm delivered ultrahigh reversible capacity (327 mAh g^−1^ at 0.1 A g^−1^), and excellent rate (146 mAh g^−1^ at 20 A g^−1^) and cycling behavior (147 mAh g^−1^ after 6000 cycles at 10 A g^−1^; 122 mAh g^−1^ after 10,000 cycles at 15 A g^−1^), indicating superior performance as compared with commercial V_2_O_5_, when applied for aqueous zinc ion batteries [22]. In contrast with aqueous zinc ion batteries, there are more recent reports on aqueous lithium ion batteries (i.e., mesoporous nanoflowers [23] and triple hollow shell [24] V_2_O_5_ structures), giving motivation to researchers to work more intensively on multivalent systems.

#### 3.1.2. Structural Stability Enhancement

The cycling stability of the cathode is affected by irreversible phase changes due to structural degradation. It has been proved that preintercalation of guest species such as Li ions, water molecules, and organic molecules can stabilize the host structure and enhance the ions’ diffusion [24,25]. In particular, Li_x_V_2_O_5_·nH_2_O had a larger spacing of 13.77 A and higher diffusion coefficient of Zn^2+^ than in V_2_O_5_·nH_2_O [26]. In that case, the enlarged layer spacing and fast Zn^2+^ diffusion favors the rate capacity and cycling performance, reaching 192 mAh g^−1^ after 1000 cycles at 10 A g^−1^.

#### 3.1.3. Surface Coating

The surface coating acts as a protective layer on the electrode surface, which can prevent volume changes and suppress the dissolution of active materials. In that case, graphene is a promising candidate to improve the electronic conductivity and inhibit dissolution. The exceptional conductivity, abundant adsorption sites, and short diffusion paths are expected to enhance the transport processes within the oxide nanomaterials. A major advantage of graphene over other carbon materials, such as graphite and carbon nanotubes, is the presence of oxygen-containing groups on the edges and surfaces, which strongly influence the size, shape, and distribution of metal oxide nanostructures on graphene.

#### 3.1.4. Formation of Solid Electrolyte Interface Film

Solid electrolyte interface (SEI) has been proved to effectively prevent the dissolution of electrode materials [27]. For instance, Zhou et al. discovered that CaSO_4_·2H_2_O protects manganese from dissolution in the case of Ca_2_MnO_4_, decreases impedance, lowers activation energy, and facilitates the intercalation/deintercalation of zinc ions without fluctuations of discharge capacity after 1000 cycles at 1 A g^−1^ [28]. This approach could also be applied for V_2_O_5_ cathode.

## 4. Multivalent Ion Properties

### 4.1. Magnesium Ion Intercalation Properties

The strong electrostatic force between Mg^2+^ and V_2_O_5_ is one of the major challenges for V_2_O_5_ as cathode material for magnesium ion batteries. An efficient approach to solve this problem is the co-intercalation of water. For instance, in the case of aqueous MgCl_2_ electrolyte solution, a high magnesium storage capacity of 427 mAh g^−1^ was estimated for aerosol-assisted chemical-vapor-deposited (AACVD) V_2_O_5_ at 500 °C [29]. Further increasing the substrate temperature to 600 °C, the capacity retention was increased to 92% after 10,000 scans, with coulombic efficiency of 100%, noble structural stability, and high reversibility [30] (Figure 1a). Studying this result, one needs to highlight the significance of AACVD to grow strongly adhered V_2_O_5_ coatings with good conformal coverage, optimized materials, and inexpensive up-scale. In addition, the temperature increase resulted in the co-existence of α-V_2_O_5_ and β-V_2_O_5_ with a compact structure of well-distributed grains (Figure 1b) as compared with α-V_2_O_5_ grown at 500 °C, which was the reason for the electrochemical enhancement. This cathode needs to be further tested in a cell to evaluate its respective performance. Sa et al. studied the magnesium storage performance of V_2_O_5_ in 1 M, magnesium bis(trifluoromethane sulfonyl)imide electrolyte, varying the water content [31]. It was found that with 2600 ppm water in electrolyte, the V_2_O_5_ cathode achieved a capacity of approximately 260 mAh g^−1^. Nevertheless, there was no evidence for reversible Mg plating/stripping limiting the use of Mg metal anode. In this work, it was also indicated that the proton involvement in the capacity value is of vital importance and should be considered with caution for future studies of possible cathode materials.

### 4.2. Zinc Ion Intercalation Properties

V_2_O_5_ was also investigated in aqueous zinc ion batteries. In this direction, Hu et al. developed an aqueous hybrid-ion battery using a mixed Zn ions/Li ions electrolyte and metallic Zn as anode [32]. In this work, it was shown that both ions can be intercalated into the porous V_2_O_5_ structure, but with a dominance of Li ions. Such a battery reached a discharge capacity of 238 mAh g^−1^ and capacity retention of 80% for 2000 cycles. This study presented the employment of “water-in-salt” electrolyte to effectively boost the energy density and the cyclability of the system. Layered Mg^2+^-intercalated V_2_O_5_ was reported to deliver high capacities of 353 and 264 mAh g^−1^ at 100 and 1000 mA g^−1^, along with long-term durability utilizing Zn foil as anode. The large radius of hydrated Mg^2+^ resulted in an interlayer spacing as large as 13.4 Å, which allowed efficient zinc ion intercalation/deintercalation [33]. Porous V_2_O_5_ nanofibers (Figure 2a) with high Zn-storage performance in an aqueous electrolyte of Zn(CF_3_SO_3_)_2_ and Zn foil as anode were reported in [34], enabling a high reversible capacity of 319 mAh g^−1^ at 20 mA g^−1^ (Figure 2b) and a capacity retention of 81% over 500 cycles. This work proposed a phase transition from orthorhombic V_2_O_5_ to zinc pyrovanadate on the first cycle, resulting in open-structured hosts enabling the particular performance.

It was also shown that the pre-intercalation of transition metal ions such as Fe^2+^, Co^2+^, Ni^2+^, Mn^2+^, Zn^2+^, and Cu^2+^ into the interlayer of V_2_O_5_ with metallic Zn as anode exhibited improved capacities, as well as excellent rate capability and cyclic stability in a temperature range of 0–50 °C. This enhancement is due to the enlarged interlayer spacing and electrical conductivity [35]. Another interesting work is related to V_2_O_5_ nanopaper, consisting of V_2_O_5_ nanofibers and carbon nanotubes, along with Zn metal pellet with a thickness of 0.35 mm as the anode, as reversible Zn ion batteries, with a high capacity of 375 mAh g^−1^ and long cycle life up to 500 cycles [36]. The enhanced behavior is due to the spaces between the oxide layers, which serve as diffusion pathways for Zn ions. In addition, the nanofiber morphology provided a short diffusion distance tolerating the high volume change. Qin et al. reported that V_2_O_5_ hollow spheres displayed a high specific discharge capacity of 132 mAh g^−1^, with capacity retention of 82.5% after 6200 cycles at the current density of 10 A g^−1^ utilizing Zn metal anode [37]. This performance is better than the one reported for commercial V_2_O_5_ due to the hollow sphere structure and the high surface area that can buffer the volume expansion of V_2_O_5_ and improve the cycling behavior of the material. The as-fabricated V_2_O_5_ @PEDOT/CC electrode displayed a maximum capacity of 360 mAh g^−1^ at 0.1 A g^−1^ and a high rate capability, with a specific capacity of 232 mAh g^−1^ at a large current density of 20 A g^−1^ utilizing a Zn foil of 0.8 mm [38]. This storage performance resulted from the synergistic effects of the V_2_O_5_ nanosheet arrays providing enough Zn storage active sites and the PEDOT coating shell increasing Zn ion/electron transport kinetics, which further acted as a protective layer to restrain structural collapse during cycling. Wang et al. reported 2D amorphous V_2_O_5_/graphene heterostructures with high capacity of 447 mAh g^−1^ at 0.3 A g^−1^, high rate capability of 202 mAh g^−1^ at 30 A g^−1^, and excellent cycle life of 20,000 cycles at 30 A g^−1^ utilizing Zn foil as anode, due to the unique features and the synergistic effect of the materials [39].

### 4.3. Aluminium Ion Intercalation Properties

The first stable aluminum ion batteries based on V_2_O_5_ nanowires with extended life were reported in 2011 by Jayaprakash et al., exhibiting a discharge capacity of 305 mAh g^−1^ in the 1st cycle and 273 mAh g^−1^ in the 20th cycle at 125 mA g^−1^ utilizing Al metal as anode [40]. The energy density of this first aluminum ion battery began the search for materials that would result in sustained improvements. Following this effort, amorphous V_2_O_5_/C composite was synthesized as a cathode for aluminum ion batteries, utilizing 250 mg of vanadium powder that was dissolved into 25 mL of 30 wt.% hydrogen peroxide solution in an ice bath, followed by the addition of 750 mg Ketjen black in the solution [41]. The morphological analysis indicated that the spherical Ketjen black particle surface was uniformly covered by V_2_O_5_. The composite was measured in glass cells with Mo plate as the current collector [41]. At a current density of 11 mA g^−1^, the composite exhibited a discharge capacity of 200 mA g^−1^ in the first cycle and about 70 mAh g^−1^ in the 30th cycle. The reversible capacity of xerogel V_2_O_5_ was estimated to be 120 mAh g^−1^ [42], with higher capacity values at lower scan rates. The loss of crystallinity occurring during the electrochemical intercalation enhanced the chemical exchange between the electrode material and the electrolyte solution.

Recently, the reversible proton intercalation/deintercalation chemistry in aqueous aluminum electrochemical cells was investigated for orthorhombic V_2_O_5_ cathode and Al anode exhibiting a capacity of approximately 200 mAh g^−1^ with high reversibility over 50 cycles. In this work, the cathode coatings composed of cation selective membranes prevented anion cross-over in aqueous electrolytes, enhancing the cathode reversibility [43].

## 5. Conclusions

In this mini review, we presented strategic approaches to provide cathode V_2_O_5_ materials of unique morphologies and crystal structures with high electrochemical performance for aqueous multivalent ion batteries. The development of V_2_O_5_ cathodes for Mg-, Zn-, and Al-ion batteries has been addressed, indicating that there are still challenges to overcome.

First, the incorporation of graphene is known to optimize the electrical conductivity of electrode materials. Nevertheless, there is no study reported on the effects of surface modification on the electrochemical properties of V_2_O_5_ for aqueous magnesium ion batteries. It will be interesting to see if the interfacial properties of the cathode can be enhanced as a consequence of the Mg^2+^ diffusion.

Second, the superconcentration strategy would be useful to apply to the expansion of the electrochemical stability window to 2.0 V and to examine if the energy density can be increased, keeping a high rate capability more intensively in all multivalent systems.

Third, the key challenge for the optimization and discovery of new materials and structures, we believe, lies in the definition of the desired performance goals to find the structure, composition, and the electrode–electrolyte interfaces that best fulfill these targets, without a priori defining the starting materials.

Forth, finding a growth process with excellent control over the structure and morphology with up-scaling capabilities is an issue that needs to be resolved for applications in the electromotive field. There is little work on CVD, but more work needs to be done. Limitation for scaling up results can be overcome through computational fluid dynamics studies of materials’ properties in numerous tests and environments to maintain optimum performance of the cathode per surface area.

Finally, another matter for consideration is the maintenance of the electrodes’ performance after the assembly and during the function of aqueous batteries under real conditions. In this case, their potential at the prototype level will initially need to be proved, and then the feasibility of their up-scaling to the industrial process level will need to be assessed. This information will be useful for the necessary steps required between the creation of innovative materials and the design of the battery.

## Figures and Tables

**Figure 1 materials-14-02310-f001:**
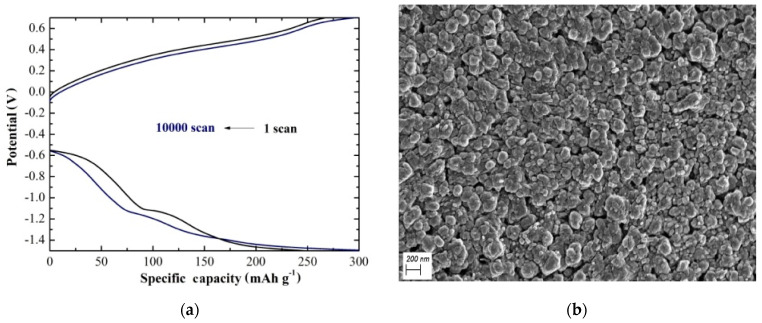
(**a**) Specific capacity for the AACVD V_2_O_5_ at 600 °C at a potential ranging from −1.5 V to +0.7 V under constant specific current of 15 A g^−1^ for 1 and 10,000 scans. (**b**) FE-SEM image of the same material at ×50,000 magnification.

**Figure 2 materials-14-02310-f002:**
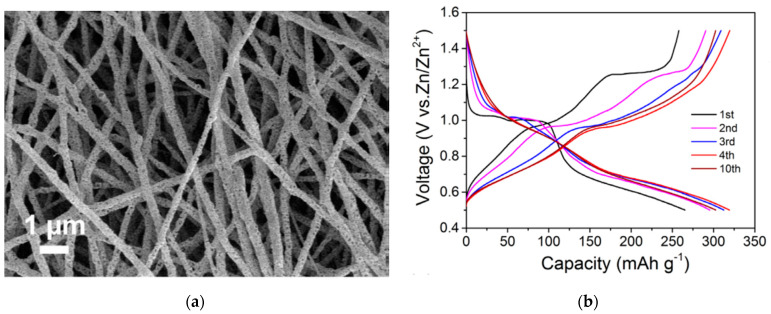
(**a**) SEM image of as-prepared V_2_O_5_ nanofibers. (**b**) Galvanostatic discharge/charge profiles at 20 mA g^−1^ [34].

## Data Availability

MDPI Research Data Policies.

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
