# Peer review of "Progress on V2O5 Cathodes for Multivalent Aqueous Batteries"

_materials, 2021, doi:10.3390/ma14092310_

Round 1

Reviewer 1 Report

The paper is not recommended for publication in its present form for the following reasons:

  1. The abstract is very general, and there is a lot of irrelevant information. The abstract should be explained and showed the important aspects of the manuscript. So, this abstract in the present form is unacceptable.
  2. The previous work is medium level, and isn't sufficient. It is recommended updated this section with new references, and compared them with your paper.
  3. As the data analysis of this paper, more statistical analysis (CV, standard deviation, mean) should been added and discussed. This lack of statistics is not acceptable in research publications and needs to be completed.
  4. In comparison to similar reviews in this topic, and some are recent and cited by the authors, it is hard to understand the contribution of this review.
  5. The authors should provide developments in this field over decades in the form of a chart or a table.
  6. The cited reference should be discussed explicitly and efficiently. Citations such as [3-7] in line 43 are not acceptable.
  7. There are too many short paragraphs with 1-2 sentences. The structure of the paper should be improved with the transition between sections and paragraphs.
  8. The critical analysis must be performed, not just the compilation of the papers and results, where is possible.
  9. The conclusion section should highlight key findings with sufficient explanations. Future works should also be briefly discussed.
  10. What are the important conclusions from the current work? What are the contribution and significance of the current work? Those questions need to be explicitly discussed.
  11. What is the purpose of writing a "mini review"? 

Author Response

Dear Reviewer 1,

Thank you very much for your response and helpful suggestions regarding our manuscript entitled:

 “Progress on V2O5 Cathodes for Multivalent Aqueous Batteries”

we wish to publish in Materials under the Special Issue of Feature Papers in Energy Materials. We have done major revisions in our original manuscript and we are resubmitting our work hoping that we have fully complied with your recommendations. The highlighted revision encloses all changes in bold and underline.

Reviewer’s comments:

The paper is not recommended for publication in its present form for the following reasons.

Thank you for the comment. Please see below our response to the points you raised.

The abstract is very general, and there is a lot of irrelevant information. The abstract should be explained and showed the important aspects of the manuscript. So, this abstract in the present form is unacceptable.

Thank you for the comment. We have proceeded in major changes of the Abstract in the revised manuscript.

The previous work is medium level, and isn't sufficient. It is recommended updated this section with new references, and compared them with your paper.

Thank you for the comment. We would initially like to clarify that our review paper contains information of the last five years and we apologise for not stating this on the first version of our manuscript.

We have replaced references that were not published the last five years and proceeded with further discussion in the whole manuscript.

The references in our work are from Journals with high impact factor related with Energy and Materials (i.e. Electrochimica Acta, Advanced Materials Interfaces, Energy Storage Materials, Nanoenergy, Journal of Energy Chemistry, Journal of Power Sources, etc.).

As the data analysis of this paper, more statistical analysis (CV, standard deviation, mean) should been added and discussed. This lack of statistics is not acceptable in research publications and needs to be completed.

Thank you for the comment. As mentioned earlier, the review paper contains research work on V2O5 cathodes for multivalent aqueous batteries. After a thorough check of our manuscript, we removed information related with other cathode materials or battery systems indicating that there is not much work done on these systems with the particular material.

Our aim was the presentation of these recent developments and the perspectives about future directions of the cathodes.

In comparison to similar reviews in this topic, and some are recent and cited by the authors, it is hard to understand the contribution of this review.

Thank you for the comment. As mentioned earlier, the focus of our paper was the presentation of the latest developments related with V2O5 cathodes for aqueous multivalent ion systems, which has not been reported to the best of our knowledge.

The authors should provide developments in this field over decades in the form of a chart or a table.

Thank you for the comment. The aim of our paper was the presentation of the developments in the last five years.

The cited reference should be discussed explicitly and efficiently. Citations such as [3-7] in line 43 are not acceptable.

Thank you for the comment. The references have been replaced and the text in the whole manuscript has been discussed further.

There are too many short paragraphs with 1-2 sentences. The structure of the paper should be improved with the transition between sections and paragraphs.

Thank you for the comment. A thorough improvement of the manuscript has been accomplished.

The critical analysis must be performed, not just the compilation of the papers and results, where is possible.

Thank you for the comment. We agree with the reviewer. Please see our response in point 7.

The conclusion section should highlight key findings with sufficient explanations. Future works should also be briefly discussed.

Thank you for the comment. The Conclusion part has been modified extensively.

What are the important conclusions from the current work? What are the contribution and significance of the current work? Those questions need to be explicitly discussed.

Thank you for the comment. Please see our response in point 9.

What is the purpose of writing a "mini review"? 

Thank you for the comment. We believe that the purpose of writing a “mini review” is the focus on a specific research topic and time period.

With our best regards,

Dr. Dimitra Vernardou

Assistant Professor

Department of Electrical & Computer Engineering

Hellenic Mediterranean University

Estavromenos, 71410 Heraklion,

Crete, Greece

Tel.: +30 2810 379631 (Office) / +30 2810 379753 (Lab)

Web: https://cuttematerials.hmu.gr/

Reviewer 2 Report

Progress on V2O5 Cathodes for Multivalent Aqueous Batteries is very interesting short review paper. Some improvement is required:

Can you explain in introduction what is main reason to give an advantage to V2O5 as cathodic material in comparison to LiCoO2 (LCO) ans LiMn2O4. Is production of V2O5 very simple and cheap?

Line 126: Nanostructured electrode materials (which particle size?)

Line 137: V2O5 nanospheres (particle size? ) prepared by ...….method,

Line 226: Mg Cl2, please to write MgCl2

Line 125: Nanostructure engineering (Which production method?)

Line 290: V2O5/C composite was synthesized ...(which method)? What is structure of V2O5/C composite (core/shell)?

Line 308, 309: What is necessary for cathode materials with high electrochemical performance for aqueous multivalent ion batteries (nanosized particles? high cristallinity? core shell structure?)

Conclusion:

Please to insert an  information about the chosen optimal method for production of V2O5, prepared structure (particle size, morphology) and obtained maximal properties regarding to capacity and  reversibility.

Author Response

Dear Reviewer 2,

Thank you very much for your response and helpful suggestions regarding our manuscript entitled:

 “Progress on V2O5 Cathodes for Multivalent Aqueous Batteries”

we wish to publish in Materials under the Special Issue of Feature Papers in Energy Materials. We have done major revisions in our original manuscript and we are resubmitting our work hoping that we have fully complied with your recommendations. The highlighted revision encloses all changes in bold and underline.

Reviewer’s comments:

Progress on V2O5 Cathodes for Multivalent Aqueous Batteries is very interesting short review paper. Some improvement is required.

Thank you for your comments. Please see below our response to all points.

Can you explain in introduction what is main reason to give an advantage to V2O5 as cathodic material in comparison to LiCoO2 (LCO) ans LiMn2O4? Is production of V2O5 very simple and cheap?

Response: Thank you for the comment. The required information has been added at the end of Introduction to justify the choice of V2O5.

Line 126: Nanostructured electrode materials (which particle size?)

Response: Thank you for the comment. The required information has been added in the revised manuscript.

Line 137: V2O5 nanospheres (particle size? ) prepared by ...….method,

Response: Thank you for the comment. The required information has been added in the revised manuscript.

Line 226: Mg Cl2, please to write MgCl2

Response: Thank you for the comment. The chemical formula has been corrected accordingly in the revised manuscript.

Line 125: Nanostructure engineering (Which production method?)

Response: Thank you for the comment. Nanostructure engineering is an approach that deals with all aspects of design and structures on the nanoscale to make useful materials and devices. The respective information has been added in the revised manuscript to clarify this point.

Line 290: V2O5/C composite was synthesized ...(which method)? What is structure of V2O5/C composite (core/shell)?

Response: Thank you for the comment. The synthesis process and the structural characteristics of the composite material have been added in the revised manuscript.

Line 308, 309: What is necessary for cathode materials with high electrochemical performance for aqueous multivalent ion batteries (nanosized particles? high cristallinity? core shell structure?)

Response: Thank you for the comment. We have added the required information in Conclusions of the revised manuscript.

Conclusion:

Please to insert information about the chosen optimal method for production of V2O5, prepared structure (particle size, morphology) and obtained maximal properties regarding to capacity and reversibility.

Response: Thank you for the comment. We believe that there is not an ideal growth process for the production of V2O5 with optimum characteristics because there are issues to overcome in each case including the electrodes assembly in a cell (i.e. in many cases it has not been tested), stability, capacity value, up-scaling, etc.). Hence, we extended the Conclusions part regarding the main challenges of materials growth and how they can possibly be handled for future work.

With our best regards,

Dr. Dimitra Vernardou

Assistant Professor

Department of Electrical & Computer Engineering

Hellenic Mediterranean University

Estavromenos, 71410 Heraklion,

Crete, Greece

Tel.: +30 2810 379631 (Office) / +30 2810 379753 (Lab)

Web: https://cuttematerials.hmu.gr/

Reviewer 3 Report

This mini-review paper focuses on vanadium oxide studied as an electrode material for aqueous magnesium-, zinc-, and aluminum-ion batteries. The manuscript should be significantly revised because it does not provide sufficient information or perspectives regarding the subject. My comments are as follows:

Multivalent-ion-based batteries have the advantage of using the corresponding metal as the anode. However, the aqueous batteries cannot use the metal as the anode, except for the zinc case. Limitations of aqueous systems, particularly lack of suitable anode materials and low cell voltage, should be commented/discussed for readers to have balanced perspectives.

Since it is a V2O5-focused review, the introduction of its crystal structure would be helpful.

Why is γ-LiV2O5 described in the main text, not V2O5? Section 3.1.2 is all about it.

Regarding section 3.1.5, V6O13 can be understood as another compound among many vanadium oxides, not just a defect structure of V2O5. If so, this section is not adequate for the V2O5 review.

Regarding the performance of the materials, capacities and rate performance are mostly described, not average discharge voltages that are also essential properties of a material. Particularly for Mg and Al cases, the anode materials used for a test cell should also be mentioned because Mg or Al metal itself might not have been used in the aqueous cells.

Calcium-ion-based batteries are also one of the multivalent-ion batteries. It should also be added to the manuscript.

Author Response

Dear Reviewer 3,

Thank you very much for your response and helpful suggestions regarding our manuscript entitled:

 “Progress on V2O5 Cathodes for Multivalent Aqueous Batteries”

we wish to publish in Materials under the Special Issue of Feature Papers in Energy Materials. We have done major revisions in our original manuscript and we are resubmitting our work hoping that we have fully complied with your recommendations. The highlighted revision encloses all changes in bold and underline.

Reviewer’s comments:

This mini-review paper focuses on vanadium oxide studied as an electrode material for aqueous magnesium-, zinc-, and aluminum-ion batteries. The manuscript should be significantly revised because it does not provide sufficient information or perspectives regarding the subject. My comments are as follows.

Thank you for your comments. Please see below our response to all points.

Multivalent-ion-based batteries have the advantage of using the corresponding metal as the anode. However, the aqueous batteries cannot use the metal as the anode, except for the zinc case. Limitations of aqueous systems, particularly lack of suitable anode materials and low cell voltage, should be commented/discussed for readers to have balanced perspectives.

Thank you for your comment. The required information has been added in Introduction for the better understanding of the particular points.

Since it is a V2O5-focused review, the introduction of its crystal structure would be helpful.

Thank you for your comment. The required information has been added in Introduction of the revised manuscript.

Why is γ-LiV2O5 described in the main text, not V2O5? Section 3.1.2 is all about it.

Thank you for your comment. We mentioned the γ-LiV2O5 because the preintercalation of Li ions was found to provide structural stability enhancement. We have removed the discussion related with Li ions in the sections 3.1.1. and 3.1.2. in the revised manuscript to avoid any confusion.

Regarding section 3.1.5, V6O13 can be understood as another compound among many vanadium oxides, not just a defect structure of V2O5. If so, this section is not adequate for the V2O5 review.

Thank you for your comment. We agree with the reviewer. The section 3.1.5 has been removed.

Regarding the performance of the materials, capacities and rate performance are mostly described, not average discharge voltages that are also essential properties of a material. Particularly for Mg and Al cases, the anode materials used for a test cell should also be mentioned because Mg or Al metal itself might not have been used in the aqueous cells.

Thank you for your comment. In the revised manuscript, we included information related with the anode material in the case the cathode was tested in a cell.

Calcium-ion-based batteries are also one of the multivalent-ion batteries. It should also be added to the manuscript.

Thank you for your comment. We decided to give emphasis on Mg-, Zn- and Al-ion aqueous batteries because they are the most recently studied.

With our best regards,

Dr. Dimitra Vernardou

Assistant Professor

Department of Electrical & Computer Engineering

Hellenic Mediterranean University

Estavromenos, 71410 Heraklion,

Crete, Greece

Tel.: +30 2810 379631 (Office) / +30 2810 379753 (Lab)

Web: https://cuttematerials.hmu.gr/

Reviewer 4 Report

Authors review paper presents the progress on vanadium pentoxides cathodes for multivalent aqueous batteries. This manuscript is possible to be published in this journal but I would like to provide several my observations and corrections :

1.  It seems to me that expressions like "good porosity" (p. 4, line 129), "chemical intercalation" (p. 5, line 166) should be avoided. They have no sense and are no clear. Also it is difficult to understand how "incorporation .... enhances the interfacial properties" (Page 6, lines 230-231).

  1. On figure 1 it is hard to see the curves of current density vs potential and potential vs specific capacity. The figures should be re-conceived and re-structured in a multi-panel form.
  2. Page 7, for "high discharge capacity", 4 different values are given 132 ( line 271), 238 (line 247), 319 (line 257), and 375 (line 270). The values differ threefold, but they are all high. Why ?
  3. In the paragraphs 3.1.1 and 3.1.2 the lithium-ion batteries are discussed which does not match the title of the manuscript.
  4. Page 5, line 185. ideal solid electrolyte interface must be non-conductive for electrons. It is strange to read SEI enhances the passage of electrons in LIB.  

In my opinion that the text of the manuscript needs several correction before publication.

Author Response

Dear Reviewer 4,

Thank you very much for your response and helpful suggestions regarding our manuscript entitled:

 “Progress on V2O5 Cathodes for Multivalent Aqueous Batteries”

we wish to publish in Materials under the Special Issue of Feature Papers in Energy Materials. We have done major revisions in our original manuscript and we are resubmitting our work hoping that we have fully complied with your recommendations. The highlighted revision encloses all changes in bold and underline.

Reviewer’s comments:

Authors review paper presents the progress on vanadium pentoxides cathodes for multivalent aqueous batteries. This manuscript is possible to be published in this journal but I would like to provide several my observations and corrections.

Thank you for the comments. Please see below our response to all points.

  1. It seems to me that expressions like "good porosity" (p. 4, line 129), "chemical intercalation" (p. 5, line 166) should be avoided. They have no sense and are no clear. Also it is difficult to understand how "incorporation .... enhances the interfacial properties" (Page 6, lines 230-231).

Thank you for the comment. The expressions “good porosity” and “chemical intercalation” have been deleted.

Regarding the sentence “incorporation .... enhances the interfacial properties" has been rephrased to clarify this point.

  1. On figure 1 it is hard to see the curves of current density vs potential and potential vs specific capacity. The figures should be re-conceived and re-structured in a multi-panel form.

Thank you for the comment. Figure 1 was re-structured in a multi-panel form and the discussion was changed accordingly.

  1. Page 7, for "high discharge capacity", 4 different values are given 132 ( line 271), 238 (line 247), 319 (line 257), and 375 (line 270). The values differ threefold, but they are all high. Why?

Thank you for the comment. The expression “high discharge capacity” was rephrased to avoid any confusion.

  1. In the paragraphs 3.1.1 and 3.1.2 the lithium-ion batteries are discussed which does not match the title of the manuscript.

Thank you for the comment. In the revised manuscript, the discussion related with Li ions in the sections 3.1.1 and 3.1.2 has been removed to avoid any confusion.

  1. Page 5, line 185. ideal solid electrolyte interface must be non-conductive for electrons. It is strange to read SEI enhances the passage of electrons in LIB.  

Thank you for the comment. We have removed the particular point to avoid any confusion.

With our best regards,

Dr. Dimitra Vernardou

Assistant Professor

Department of Electrical & Computer Engineering

Hellenic Mediterranean University

Estavromenos, 71410 Heraklion,

Crete, Greece

Tel.: +30 2810 379631 (Office) / +30 2810 379753 (Lab)

Web: https://cuttematerials.hmu.gr/

Round 2

Reviewer 1 Report

The revised form of the paper is good.

Reviewer 3 Report

Many of the reviewer's comments were addressed properly and the manuscript has been significantly improved. I recommend its publication in this journal.